# Paracrine Interaction of Cholangiocellular Carcinoma with Cancer-Associated Fibroblasts and Schwann Cells Impact Cell Migration

**DOI:** 10.3390/jcm11102785

**Published:** 2022-05-15

**Authors:** Jan-Paul Gundlach, Jannik Kerber, Alexander Hendricks, Alexander Bernsmeier, Christine Halske, Christian Röder, Thomas Becker, Christoph Röcken, Felix Braun, Susanne Sebens, Nils Heits

**Affiliations:** 1Department of General, Visceral-, Thoracic-, Transplantation- and Pediatric Surgery, University Medical Center Schleswig-Holstein (UKSH), Campus Kiel, Arnold-Heller-Str. 3, Building C, 24105 Kiel, Germany; jannik-kerber@t-online.de (J.K.); alexander.bernsmeier@uksh.de (A.B.); thomas.becker@uksh.de (T.B.); felix.braun@uksh.de (F.B.); nheits@web.de (N.H.); 2Institute for Experimental Cancer Research, Kiel University and University Medical Center Schleswig-Holstein (UKSH), Campus Kiel, Arnold-Heller-Str. 3, Building U30, 24105 Kiel, Germany; christian.Roeder@uksh.de (C.R.); susanne.sebens@email.uni-kiel.de (S.S.); 3Department of General, Visceral-, Vascular-, and Transplantation Surgery, Medical University Rostock, Schillingallee 35, 18057 Rostock, Germany; alexander.hendricks@med.uni-rostock.de; 4Institute of Pathology, University Medical Center Schleswig-Holstein (UKSH), Campus Kiel, Arnold-Heller-Str. 3, Building U33, 24105 Kiel, Germany; christine.halske@uksh.de (C.H.); christoph.roecken@uksh.de (C.R.); 5Gesundheitszentrum Kiel-Mitte, Prüner Gang 15, 24103 Kiel, Germany

**Keywords:** cholangiocellular carcinoma, cancer-associated fibroblasts, Schwann cells, tumor stroma, CCA, Sorafenib

## Abstract

Although the Mitogen-activated protein kinase (MAPK) pathway is enriched in cholangiocarcinoma (CCA), treatment with the multityrosine kinase-inhibitor Sorafenib is disappointing. While cancer-associated fibroblasts (CAF) are known to contribute to treatment resistance in CCA, knowledge is lacking for Schwann cells (SC). We investigated the impact of stromal cells on CCA cells and whether this is affected by Sorafenib. Immunohistochemistry revealed elevated expression of CAF and SC markers significantly correlating with reduced tumor-free survival. In co-culture with CAF, CCA cells mostly migrated, which could be diminished by Sorafenib, while in SC co-cultures, SC predominantly migrated towards CCA cells, unaffected by Sorafenib. Moreover, increased secretion of pro-inflammatory cytokines MCP-1, CXCL-1, IL-6 and IL-8 was determined in CAF mono- and co-cultures, which could be reduced by Sorafenib. Corresponding to migration results, an increased expression of phospho-AKT was measured in CAF co-cultured HuCCT-1 cells, although was unaffected by Sorafenib. Intriguingly, CAF co-cultured TFK-1 cells showed increased activation of STAT3, JNK, ERK and AKT pathways, which was partly reduced by Sorafenib. This study indicates that CAF and SC differentially impact CCA cells and Sorafenib partially reverts these stroma-mediated effects. These findings contribute to a better understanding of the paracrine interplay of CAF and SC with CCA cells.

## 1. Introduction

Cholangiocarcinoma (CCA) is the most common malignancy of the biliary tract and the second most common primary hepatic malignancy [1]. Its incidence in Western countries is increasing [2], which particularly applies to intrahepatic CCA [3]. Treatment options for CCA are limited. Until now, the only curative therapy is resection of the primary tumor. Cisplatin and Gemcitabine are used as standard practice chemotherapeutics [4], though due to a lack of qualified data, a lively debate on well-established and broadly accepted therapeutic regime is ongoing [5]. Furthermore, CCA patients may benefit from concomitant radiotherapy, and also liver transplantation can be an option in selected cases [5]. Other current treatment regimens for CCA patients imply transarterial chemoembolization (TACE), intra-arterial chemotherapy and radiofrequency ablation (RFA) approaches [1,5].

Whereas targeted therapies have become effective therapeutic options in different tumor entities, biological treatments for CCA are still scarce [6]. Genetic and microarray data analysis revealed that genes of the Mitogen-activated protein kinase (MAPK) pathway are enriched in CCA as similarly seen in hepatocellular carcinoma (HCC) [7] and the Mitogen-Activated Protein Kinase (MAPK) pathway was demonstrated to be activated in CCA tissue compared to non-tumorous tissue [8]. Since, in advanced HCC, the multityrosine kinase inhibitor Sorafenib was the first drug to significantly improve overall survival [9], Sorafenib was proposed to be advantageous in CCA treatment too [7]. In vitro studies with Sorafenib treatment revealed contrasting results [10,11]: Huether et al. were able to demonstrate the inhibition of cell growth in CCA cell lines by showing that a combined treatment of Sorafenib with Doxorubicin or insulin-like growth factor-1 receptor (IGF-1R)-inhibition resulted in additive antiproliferative effects. However, co-application of Sorafenib and the antimetabolites 5-Fluoruracil (5-FU) or Gemcitabine diminished the antineoplastic effects of cytostatic drugs [10]. Overall, these data indicate that Sorafenib may exert its effects in a context-dependent manner.

As in other solid tumors, CCA cells are embedded in a tumor microenvironment consisting of extracellular matrix (ECM) and a variety of inflammatory stromal cells. Thus, tumor and stromal cells reciprocally interact and impact each other, thereby essentially driving malignant progression and also affecting therapy responses [12]. Interestingly, apart from the MAPK pathway, the above-mentioned genetic analysis also revealed genes encoding for ECM proteins to be significantly enriched [7]. In the tumor stroma, myofibroblasts, also termed cancer-associated fibroblasts (CAF), are of particular interest [13] because they often represent one major stroma cell population. CAF are characterized—amongst others—by high expression of α-smooth muscle actin (α-SMA) and fibroblast specific protein-1 (FSP-1) and produce a plethora of inflammatory mediators, growth factors and ECM molecules by which they promote malignant progression of different tumor entities [14,15]. Even though a high abundance of CAF was correlated with poor survival of CCA patients [16] and CAF are known to contribute to treatment resistance in CCA [13], detailed analyses on the impact of CAF on CCA cells are still missing.

Apart from CAF, Schwann cells (SC) are also an important part of the tumor microenvironment of CCA, though their contribution to tumor progression is not yet fully understood [17]. SC represent a major component of the peripheral nervous system in their function in myelination, axonal maintenance and repair [17]. SC are commonly detected in tumor tissues by staining of the protein S100, which is a calcium binding protein [18]. Several studies indicate that nerves have a stimulatory role in cancer progression [19,20,21], e.g., in promoting tumor cell (perineural) invasion [21].

Since knowledge on the role of CAF and SC in CCA progression is still very poor, the present study aimed to better understand the impact of either stromal cell population on CCA migratory abilities and to analyze whether treatment with Sorafenib might be effective in inhibiting CCA migration.

## 2. Results

### 2.1. High Abundance of CAF and SC Is Associated with Reduced Survival of CCA Patients

First, tumor sections of 14 CCA patients (5 intrahepatic and 9 extrahepatic) were immunohistochemically analyzed for the abundance of CAF (by staining of α-SMA and FSP-1) and SC (by staining of S100). Of these cases, three patients were operated in TNM-stadium T1, nine patients in stadium T2 and two patients in T3. Representative stainings of CAF and SC in a T1 tumor and a T3 tumor are shown in Figure 1A, indicating that the accumulation of either stromal cell population already starts in early stage CCA. Overall, a moderate to strong staining of either α-SMA or FSP-1 was detected in 10/14 tissues as well as a weak to moderately positive S100 staining in 5/14 tissues. Moreover, elevated expression of either marker showed a reduced tumor-free survival of the patients (Figure 1C–E). Although obtained from a small cohort, these findings support previous studies [14] and underscore that a high abundance of CAF and SC is associated with reduced survival of CCA patients.

### 2.2. Impact of CAF and SC on Cell Migration of CCA Cells

To elucidate the impact of CAF and SC on the migratory behavior of CCA cells, HuCCT-1 and TFK-1 cells as well as both stromal cell populations were cultured either alone (mono-culture) or CCA cell lines were co-cultured with CAF or SC in four chamber ibidi slides. Cell migration was assessed by measuring the gap closure by determining cell confluence (Figure 2). Figure 2A–C show representative images of one experiment (out of three independent experiments). Mono-cultured CCA cell lines as well as stromal cells only led to a gap closure of 5.4% (mono-cultured CAF) to 17.5% (mono-cultured HuCCT-1) after 18 h (Figure 2D–G). Since HuCCT-1 cells showed a higher basal migratory potential than TFK-1 cells (17.5% versus 1.4% TFK-1 cells after 18 h), gap closure was analyzed after 18 h in HuCCT-1 (Figure 2D,F) and in TFK-1 cells after 18 (not shown) and 68 h (Figure 2E,G). Moreover, while gap closure by mono-cultured CAF was only marginally increased from 18 h to 68 h (5.4 to 10.5%), gap closure by mono-cultured SC increased from 6.8% after 18 h to 49.2% after 68 h, indicating a higher basal migratory potential of SC (Figure 2F,G).

Importantly, gap closure was most pronounced when either CCA cell line was co-cultured with stromal cells, albeit clear differences were observed in dependence on the stromal cells (Figure 2A,B,D–G). While in the co-culture of CCA cells and CAF, the gap was closed predominantly by migration of CCA cells (Figure 2A), gap closure during co-culture of CCA cells and SC was predominantly mediated by the migration of SC cells towards CCA cells (Figure 2B). This migratory behavior was also not changed when CCA cells were concomitantly exposed to CAF and SC (Figure 2C). In HuCCT-1 co-cultures, gap closure was significantly more pronounced in the presence of CAF (*p* < 0.0001) than SC (*p* = 0.4489; 66.9% gap closure after CAF co-culture versus 24.8% after SC co-culture, Figure 2D,F), while in TFK-1 co-cultures, gap closure was slightly more pronounced in the presence of SC after 18 h (*p* = 0.0083) but comparable in the presence of CAF (*p* = 0.0287; 44.7%) and SC (*p* = 0.0093; 50.1%) after 68 h (Figure 2E,G). Overall, these data indicate that cell migration of CCA cells is enhanced in the presence of CAF, while the presence of CCA cells, in turn, increases cell migration of SC.

### 2.3. Sorafenib Inhibits Migration of CCA and Stromal Cells in a Context Dependent Manner

Since the multityrosine kinase inhibitor Sorafenib has already shown growth inhibitory effects on CCA cells [7], we next investigated whether Sorafenib might be also able to inhibit cell migration of CCA cells and different stromal cells. Since a dosage of 1 µM has been shown to hardly impact the cell growth of neither CCA cells nor both stromal cell populations (data not shown), mono- and co-cultures of CCA cells, CAF and SC, respectively, in the absence or presence of 1 µM Sorafenib were conducted to analyze gap closure. As seen in Figure 3A, Sorafenib treatment even intensified cell migration of mono-cultured HuCCT-1 cells, leading to an increased gap closure from 17.5% to 28.2%, while cell migration of mono-cultured CAF was clearly diminished after 18 h (from 5.4% to 2.9%, *p* = 0.0668, Figure 3A,B) and even more pronounced after 68 h (from 10.5% to 3.8%, *p* = 0.0005, Figure 3C). Furthermore, a clear reduction in cell migration was observed in co-cultures from HuCCT-1 cells and CAF (from 66.9% to 38.0%, *p* = 0.0548). In contrast, gap closure by mono-cultured SC after 18 h (8.2% versus 8.6%, Figure 3A,B) and after 68 h (49.3% versus 45.2%, Figure 3C) as well as HuCCT-1 co-cultures with SC (from 24.8% to 28.5%; Figure 3A) remained nearly unaffected. Similar observations could be made with TFK-1 cells. Here, Sorafenib treatment also did not impact cell migration of mono- and co-cultured CCA cells with SC (Figure 3B,C). In line with the results described for HuCCT-1 cells and mono-cultured CAF, a clear inhibitory effect on cell migration could only be observed on co-cultures of CAF and TFK-1 cells, leading to a reduced gap closure from 8.9% to 4.5% (*p* = 0.0145) and from 44.5% to 23.4% (*p* = 0.1489) after 18 h and 68 h, respectively (Figure 3B,C). Overall, these data indicate that Sorafenib treatment impairs the migratory abilities of CAF and CCA cells but not SC. The fact that particularly the co-culture increased cell migration of CAF and CCA cells was affected point to a Sorafenib-mediated inhibition of paracrine interactions between tumor cells and CAF.

### 2.4. Analysis of Potential Migratory Inducing Factors in Co-Culture with CAF or SC

In order to better understand the paracrine interactions between CCA cells and stromal cell populations, which might impact cell migration and might be affected by Sorafenib treatment, Proteome Profiler Human Cytokine Array Kit Panel A was used to determine the release of cytokines and chemokines during the different mono- and co-cultures in the absence or presence of 1 µM Sorafenib. No differences in the release of C5a, CD40 Ligand/TNFSF5, G-CSF, GM-CSF, CXCL1/GROα, CCL1/I-309, ICAM-1, IFN-µ, Interleukin (IL)-1α /IL-1F1, IL-1ß/IL-1F2, IL-1ra/IL-1F3, IL-2, IL-4, IL-5, IL-6, IL-8, IL-10, IL-12 p70, IL-13, IL-16, IL-17, IL-17E, IL-23, IL-27, IL-32α, CXCL10/IP-10, CXCL11/I-TAC, CCL2/MCP-1, MIF, CCL3/MIP-1α, CCL4/MIP-1ß, CCL5/RANTES, CXCL12/SDF-1, serpin E1/PAI-1, TNF-α and TREM-1 could be observed in supernatants of the different co-cultures compared to respective mono-cultures as well as after treatment with Sorafenib (data not shown). In contrast, clearly detectable levels of Monocyte Chemoattractant Protein-1 (MCP-1), Chemokine (C-X-C motif) ligand 1 (CXCL-1), IL-6 and IL-8 could be determined in supernatants of CAF as well as in supernatants of co-cultures of CAF and either CCA cell line (Figure 4A). Moreover, after Sorafenib treatment, the levels of all fac tors were strongly reduced compared to the respective untreated samples (Figure 4A).

Furthermore, considerable amounts of serpin E1 and Macrophage migration inhibitory factor (MIF) could be determined in supernatants of mono-cultured HuCCT-1 and SC as well as in co-cultures of both CCA lines and SC (Figure 4B), while in supernatants of mono-cultured TFK-1 cells, no serpin E1 could be detected. However, Sorafenib treatment did not impact serpin E1 and MIF levels in supernatants of either condition. Overall, these data indicate that Sorafenib mostly inhibited the release of CAF-derived factors.

### 2.5. Altered Signal Transduction in CAF and SC Stimulated CCA Cells

To elucidate whether the paracrine interactions with stromal cells led to an altered signal transduction in CCA cells, Western blot analyses of CCA cells exposed to medium or conditioned medium from either stromal cell line in the absence or presence of 1 µM Sorafenib were conducted to detect activated (phosphorylated) forms of STAT3, JNK, ERK and AKT (Figure 5A).

In HuCCT-1 cells, expression levels of phospho-STAT3 and phospho-JNK were not altered, while increased expression of phospho-JNK and phospho-AKT could be determined when CCA cells were cultured in CAF conditioned medium. However, treatment with Sorafenib showed no effect on activation of either protein and even increased expression of phospho-ERK (*p* = 0.0084; Figure 5A,C). Intriguingly, CAF-conditioned TFK-1 cells showed increased activation of all four signaling mediators (highly significantly for phospho-STAT3 and phospho-ERK with *p* = 0.0011 and *p* = 0.0072, respectively) and Sorafenib treatment reduced it (Figure 5B,C).

HuCCT-1 cells cultured in SC-conditioned medium showed almost no changes in the expression of activated forms of the MAPK and JAK-STAT signaling pathway. Again, a slight increase in phospho-ERK expression after Sorafenib treatment could be observed (Figure 5A,C). In contrast, TFK-1 cells exposed to SC-conditioned medium exhibited an increased activation of STAT3, JNK and AKT (Figure 5B,C), whereas no SC-mediated effects could be observed on the expression level of phospho-ERK. Treatment with Sorafenib resulted in a reduced phosphorylation of JNK and AKT, but also in an increased phosphorylation of ERK in TFK-1 cells. Again, no synergistic effect of the CCA cell lines in co-cultivation with both stromal cell lines was detected on signal transduction (data not shown).

To rule out a direct inhibition of activation of the above-mentioned pathways in the CCA cells by Sorafenib, Western blot analysis was carried out in Sorafenib-treated mono-cultured TFK-1 and HuCCT-1 cells. For the TFK-1 cells, a light activation of ERK and JNK was observed after Sorafenib treatment (AUC in phospho-ERK from 201 to 8490 and in phospho-JNK from 331 to 5941). This finding in the mono-culture contrasted to the inhibition of the different pathways after Sorafenib treatment of the cancer cell-SC and -CAF co-cultures. For the AKT and JAK-STAT pathway, no difference in activation was observed after treatment with Sorafenib. In mono-cultured HuCCT-1 cells, no difference in activation of the different pathways was observed after treatment with Sorafenib either (phospho-STAT3 from 6625 to 5992; phospho-JNK from 7925 to 7627; phospho-AKT from 5278 to 2302 and phospho-ERK from 9109 to 8432 AUC, respectively).

Altogether, these findings indicate that CAF and SC impact signaling pathways in CCA cells in a paracrine manner. Moreover, Sorafenib treatment inhibited only the activation of JNK, AKT and ERK in TFK-1 cells.

## 3. Discussion

Although CCA is less frequent than other gastrointestinal malignancies, its incidence is rapidly increasing [2]. Additionally, CCA treatment remains challenging because of its often late diagnosis, limiting surgical resection, poor response to standard chemotherapy, missing standardized second-line chemotherapeutic approaches and difficulty for targeted therapies due to tumor heterogeneity without established molecular-targeted therapeutic regimens [5,22]. Thus, this underscores the urgent need to develop more effective treatment options. While the tumor microenvironment and, particularly, CAF have been already shown to essentially determine the progression and therapy response of many tumor entities [23], knowledge on the role of CAF on CCA cells and CCA progression is still poor [13]. Chuaysri et al. could already demonstrate that the occurrence of CAF is correlated with poor survival of CCA patients [16]. This finding was confirmed by the immunohistochemical analysis of a small cohort of CCA patients in our study, which requires further validation in larger cohorts. Even though the impact of SC on tumor cell migration, invasion and metastasis has been reported in different malignancies (e.g., pancreatic ductal adenocarcinoma) [24], their role in CCA progression and metastatic spread has not yet been investigated.

Since the role of CAF and SC on CCA progression is poorly understood and the effects of treatment with the multityrosine kinase inhibitor Sorafenib are rather disappointing in CCA [25,26], our study aimed at elucidating the paracrine impact of CAF and SC on CCA cells with a particular focus on cell migration and whether this can be targeted by Sorafenib.

Using four chamber ibidi slides and cell tracing of distinct cell populations for gap closure experiments, we were able to demonstrate that cell migration of CCA cells is significantly enhanced in the presence of CAF. One possible explanation could be the paracrine interaction with tumor cells as CAF are known to produce ECM and release inflammatory cytokines such as IL-6 and IL-8 and several chemokines, among them CCL2/MCP1, CXCL12/SDF1, CCL5 and 7 as well as CXCL16, by which they can promote cancer cell migration [27]. In contrast, in co-cultures of CCA cells with SC, gap closure was also increased but this was predominantly mediated by enhanced cell migration of SC towards CCA cells. These data are in line with a study demonstrating that SC migrate towards cancer cells in pancreatic ductal adenocarcinoma as well as in colon carcinoma cell lines but not towards benign cells [19]. Thus, the study by Demir et al. and our study provide evidence that not tumor cells but, instead, SC migrate during neural invasion which might contribute to tumor cell dissemination.

We were also able to demonstrate that Sorafenib could reduce migration of CAF as well as of CAF co-cultivated CCA cells but not that of SC and SC co-cultivated CCA cells, supporting the view that Sorafenib exerts its effects in a context-dependent manner.

However, the findings that Sorafenib reduced the migratory abilities of CAF and CAF co-cultured CCA cells are in line with other studies demonstrating the antimigratory effects of Sorafenib [26,27]. Accordingly, Sorafenib was found to suppress the Epithelial–Mesenchymal Transition (EMT) and cell migration by reducing matrix metalloproteinase (MMP) expression in HCC cells. This in turn led to suppressed c-MET and reduced activation of the Mitogen-activated protein kinase (MEK)/ERK pathway [28]. In line with these findings, Sorafenib reduced phospho-ERK levels and migration of breast cancer cells [29]. Considering the effects of Sorafenib in HCC, the inhibited ERK pathway highlights one possibility by which Sorafenib treatment was able to inhibit migration in CAF and CAF co-culture-treated CCA cells. SC dedifferentiation is known to be activated through the Ras/Raf/ERK signaling pathway [30] and it was demonstrated that ERK1/2 and AKT signals were involved in the migratory potential of SC [31]; however, ERK1/2 activity inhibition did not show a reduction in SC migration [31]. In line with this, migration of mono- and co-cultured SC could not be inhibited by Sorafenib. The observed resistance of SC and SC co-culture to the treatment with Sorafenib might be caused either by a shorter and weaker ERK1/2 activity in SC or a possible requirement for additional factors, such as insulin-like growth factor to stimulate the ERK1/2 pathway [32]. Future studies are needed to explore these findings more in detail.

To further elucidate the paracrine impact of CAF and SC, respectively, on CCA cells and whether this is impacted by Sorafenib treatment, we determined a spectrum of cytokines and chemokines in supernatants of mono- and co-cultures of CCA cells and the different stromal cell populations. From the series of investigated cytokines, the following have been shown to be of particular importance for tumor cell migration: IL5 [33], IL6 [34], IL8 [35], IL17 [36], IL23 [37], CXCL-1, CXCL11 [38], CXCL12 [39], CCL5 [40], MCP-1 [41], MIF [42], serpin E1 [43] and TNFα [44]. While MCP-1, CXCL-1, IL-6 and IL-8 could not be detected in supernatants of mono-cultured CCA cells, a strong release of MCP-1 and IL-8 was measured in supernatants of mono-cultured CAF as well as of CCA cells co-cultured with CAF. Both factors are known for their pro-inflammatory [45] and migratory capacity [27]. Of note, MCP-1 (also known as CCL2) has been reported to be secreted by CAF at elevated levels, promoting cancer progression and migration in HCC [46] as well as in oral cancer [47]. Additionally, CXCL-1 and IL-6 could also be detected at elevated levels in supernatants of mono-cultured CAF, both factors being reported to be highly expressed in CCA [48]. Our findings are in line with studies demonstrating a role of CXCL-1, IL-6 and IL-8 in tumor cell migration and metastasis in other tumor entities [49,50]. Importantly, Sorafenib treatment effectively reduced levels of MCP-1, CXCL-1, IL-6 and IL-8 in supernatants of mono-cultured CAF and CAF co-cultured with either CCA cell line, which is in line with the observation that Sorafenib efficiently reduced cell migration of mono-cultured CAF as well as those of CAF co-cultured CCA cells. Overall, these findings indicate that Sorafenib is able to efficiently interfere with the paracrine tumor stroma interplay in CCA.

Furthermore, supernatants of mono-cultured HuCCT-1 cells as well as of mono- and co-cultured SC contained considerably higher amounts of serpin E1 and MIF. High MIF levels are found in almost all type of cancers exerting multifunctional effects contributing to cancer development and progression such as promoting migration and reducing apoptosis [51]. In addition, it is known to inhibit the tumor suppressor gene p53 and its stimulatory function of pro-inflammatory cytokines such as TNFα, INF γ, IL-1ß, IL-6, and IL-8 in a positive feedback loop [52]. It appears possible that SC migration is enhanced by MIF. In this study, we analyzed the secretion of the cytokines in mono- and co-culture of CCA cells, SC and CAF and examined whether paracrine interaction of the above mentioned cell types and Sorafenib treatment impact this cytokine and chemokine secretion. Based on these results, it can be speculated whether enhanced factors under co-culture conditions are responsible for enhanced migratory abilities of CCA cells or SC migration. Thus, future studies involving experiments with blocking antibodies will have to examine whether these inflammatory factors (such as MCP-1, CXCL-1, IL-6 or IL-8) are responsible for enhanced migration of CCA cells and SC, respectively. As an overrepresentation of the MAPK/ERK signaling pathway was already described in CCA [7], Sorafenib has been postulated to be an effective treatment option like in HCC. Having shown that Sorafenib reduced cell migration of CAF and CAF co-cultured CCA cells, we elucidated the paracrine influence of CAF on signal transduction in the two CCA cell lines as well as whether this is impacted by Sorafenib treatment. In addition, we investigated activation of the PI3K-AKT and JAK-STAT signaling pathways, which are both known to be possible targeting structures for CCA treatment [53]. First of all, HuCCT-1 cells already showed a stroma-independent higher expression of activated signaling mediators, particularly of phospho-AKT and phospho-ERK compared to TKF-1 cells. Moreover, in HuCCT-1 cells, only phospho-AKT expression was increased in response to CAF, while the presence of SC did not impact the expression of either phosphorylated form of the analyzed signaling proteins. Interestingly, Sorafenib treatment did not alter the expression of the activated forms of the four signaling mediators in HuCCT-1 cells under either culture condition, though a clear reducing effect was observed on cell migration in CAF co-cultivated cells, possibly due to decreased levels of CXCL-1, IL-6 or IL-8 leading to reduced migration of HuCCT-1 cells. In contrast, low basal expression levels of phospho-JNK, phospho-AKT and phospho-ERK in TFK-1 cells were increased in a stroma-dependent manner by CAF, which could be clearly reduced by Sorafenib treatment.

Of note, while Sorafenib seemed to have little to no effects on cytokine levels in SC co-cultured cells, SC co-cultured TFK-1 cells showed an increased activation of ERK, which was even enhanced by Sorafenib treatment. A reason for the stronger migration of SC in co-culture with extrahepatic TFK-1 cells might be explained by the fact that extrahepatic CCA disseminate into the liver via perineural guidance [21], while intrahepatic carcinomas (such as the HuCCT-1 cells) are more surrounded by CAF and hepatic stellate cells in the tumoral environment [13]. Therefore, intrahepatic CCA (cells) might be primarily activated by a paracrine stimulation from stellate cells/CAF and less by SC. Thus, there are apparent differences in the activation and migration of SC and tumor cells, respectively, which might be caused by the different paracrine interplay of CCA cells and surrounding stromal cells. Furthermore, the antitumoral efficacy of Sorafenib seems to be dependent on the tumoral context as, predominantly, the interplay between CAF and CCA cells could be impaired by Sorafenib but not those of SC and CCA cells. Thus, it can be speculated that Sorafenib treatment might be more effective and suitable for treatment of CCA lacking neural invasion.

In line with other studies, the results of our study showed lower survival rates in CCA patients with an increased immunohistochemical staining of α-SMA in CAF in surgically resected intrahepatic CCA [16,54,55]. Prospective risk assessment regarding survival after surgery could favor patients with a low density of α-SMA in CAF. Therefore, tumor biopsies in the evaluation process before surgery could add to a better selection of patients who might benefit from surgery with a better tumor-free survival for CCA. Furthermore, the results of the study suggest that a suppression of the crosstalk between CAF and CCA tumor cells leads to an impaired tumor invasion. Therefore, future research should focus on identifying targets by which deregulated expression and release of tumor-stimulating cytokines by CAF are reversed. Of note, since CAF have been demonstrated to be heterogeneous in different tumor entities, especially in pancreatic and breast cancer [13], future research should also focus on CAF heterogeneity in CCA in order to elucidate whether different CAF subtypes might exert pro- and antitumorigenic effects in CCA cells. Moreover, owing to the fact that our results are based on the use of two CCA cell lines that differ in their origin (intrahepatic vs. extrahepatic) and therefore generalization is difficult in the presence of inconsistent findings observed, the use of additional CCA cell lines as well as CAF populations is planned for further studies. For extrahepatic CCA, a migration of SC towards the tumor cells was seen. Extrahepatic CCA cells need SC for perineural invasion to invade in the liver [21]. Since the results of this study demonstrate migration of SC towards CCA cells and a synergistic effect of SC and CCA cells in co-culture, future research should investigate the mechanisms of the migration of SC towards CCA cells in more detail and targets should be identified to block this malignancy, promoting crosstalk.

## 4. Materials and Methods

### 4.1. Immunohistochemical Stainings of CCA Tissues

Formalin-fixed and paraffin-embedded tumor specimens of 14 CCA patients were analyzed for the presence of CAF and SC and correlated with survival rates. All patients signed the informed consent. The study was approved by the local institutional review board of the Medical Faculty of the Kiel University (A 110/99). For immunohistochemistry, 5 µm paraffin sections were used. Antigen retrieval was not necessary. Slides were incubated with primary antibodies for α-SMA (1:400, clone 1A4, mouse, NeoMarkers, Fremont, CA, USA), FSP-1 (1:200, abcam ab93283, Cambridge, UK) and S100 (1:400, Z0311, polyclonal, rabbit, Dako, Glostrup, Denmark). Bound antibodies were detected by EnVision+System-HRP anti-mouse antibody (Dako, Glostrup, Denmark). Color development was performed with the DAB substrate kit (Dako, Glostrup, Denmark). All slides were counterstained with hemalum and cover slipped. Evaluation of the stainings was performed on a Leica DM 1000 microscope (Leica, Wetzlar, Germany). The intensity of the staining was judged on an arbitrary scale of 0 to 3 with 0: no staining; 1: weak staining; 2: moderate staining and 3: strong staining by two independent pathologists.

### 4.2. Cell Lines and Generation of CAF

CAF were prepared from two different human CCA tissues and pooled after in vitro selection as previously described and tested by positive staining for α-SMA and Vimentin as well as negative staining for the pan-cytokeratin marker KL-1 [56]. Patients gave their consent for use of their tumor tissue and the procedure was approved by the ethics committee of Kiel University (A 110/99). In brief, immediately after resection, the tissue was cultivated with DMEM low glucose (Gibco Invitrogen, Grand Island, NY, USA), 10% FCS (PAN Biotech, Aidenbach, Germany), 1% GlutaMAX, 1% sodium pyruvate (both Gibco Invitrogen, Grand Island, NY, USA) and 1% penicillin/streptomycin (Biochrom, Berlin, Germany), sliced into 1 mm³ pieces and cultivated in 6-well plates in DMEM low glucose medium (10% FCS, 1% GlutaMAX, 1% sodium pyruvate, as described above). After 2 days, pieces were transferred into new 6-well plates. The medium was changed three times per week. After 2 weeks, when sufficient CAF were migrated out of the tissues, the tumor blocks were removed. At a confluency of 80%, CAF were detached with accutase (Gibco Invitrogen, Grand Island, NY, USA) and the cells were stored in FCS with 10% DMSO (Sigma-Aldrich Chemie GmbH, Steinheim, Germany) in liquid nitrogen for further studies.

The human CCA cell lines HuCCT-1 and TFK-1 (Cell Bank RIKEN Bio Resource Centre, Koyadai Tsukuba, Japan) as well as CAF were cultured in DMEM low glucose medium supplemented with 10% FCS, 1 mM GlutaMAX and 1 mM sodium-pyruvate (culture medium), as described before. SC were purchased from ScienCell Research Laboratories (San Diego, CA, USA) and kept in SC culture medium of the distributor.

### 4.3. Paracrine Interaction Model to Analyze Cell Migration

To analyze the influence of SC and CAF on growth behavior and cell migration of CCA cell lines, HuCCT-1 and TFK-1 cells were resuspended in culture medium without FCS but supplemented with 25 µM CellTracker Deep Red. SC were resuspended in culture medium without FCS and supplemented with 25 µM CellTracker CMAC blue (both dyes from Molecular Probes by Life Technologies, Waltham, MA, USA). They were stained for 45 min at 37 °C and subsequently resuspended in cell culture medium containing FCS. CAF remained unlabeled. Each cell population was seeded at a cell density of 0.35 × 10^5^ cells in each well of the ibidi plate (Culture-Insert 4 well in µ-Dish, ibidi, Munich, Germany) in 75 µL DMEM low glucose with 10% FCS. For mono-culture, CCA cell lines as well as CAF and SC were seeded in all 4 wells of the ibidi dish, while for co-culture experiments, either CCA cell line was seeded in 2 wells and in the other two wells either CAF or SC were seeded. The next day, the medium was removed from all wells and replaced by 75 µL DMEM FluoroBrite high-glucose (Gibco Invitrogen) lacking FCS. In the case of Sorafenib treatment, medium was replaced by 75 µL DMEM FluoroBrite (-FCS) containing 1 µM Sorafenib. After 4 h, the stamp was removed and the ibidi plates were washed with DMEM FluoroBrite (-FCS) in order to remove detached cells or cellular debris that had arisen after removal of the stamp. The ibidi plates were then refilled with DMEM FluoroBrite (-FCS) for the control and DMEM FluoroBrite (-FCS) with 1 µM Sorafenib for the treatment. Subsequently, selected regions of the well containing the gaps were imaged in all four channels (brightfield: Ex: BF, Em: Blue (452/45); CellTracker CMAC Blue: Ex: UV (377/50), Em: Blue (452/45); CellTracker Deep Red: Ex: Red (632/22), Em: Red (685/40)) using the 10x objective of a NYONE cell imager (Synentec GmbH, Elmshorn, Germany). To analyze the cell confluence in the brightfield channel, the ‘Wound healing beta 2F’ application of YT-Software (SYNENTEC GmbH) was used. In this application, four regions of interest (ROIs) could be placed into the four gaps of each well and the confluency within this ROI was determined. To determine gap closure, cellular confluency was measured directly after removal of the stamp (=t0) as well as after 18 h and 68 h. As cellular debris or detached cells could interfere with the analysis, after 18 h and 68 h, the medium was collected, filtered (0.45 µm) and directly re-applied. After 72 h, supernatants were centrifuged, aliquoted and stored in −20 °C for further experiments (cytokine assay).

### 4.4. Human Cytokine Array

To further characterize the impact of CAF and SC on tumor cell lines, supernatants of mono- and co-cultures were analyzed after 72 h for cytokine release using the Proteome Profiler Human Cytokine Array Kit (Panel A, R&D SYSTEMS, Minneapolis, MN, USA) following the manufacturer’s specifications. For detection of cytokines, the substrate was catalyzed with peroxidase into a luminescent product, which was detected on a Curix-60 developer (AGFA, Mortsel, Belgium).

### 4.5. Generation of Conditioned Medium of Stromal Cells

Conditioned medium of mono-cultured stromal cells was prepared as follows in T75 cm² cell culture flasks: 80–90% confluent CAF or SC were cultured with 10 mL 0% FCS medium (CAF in DMEM low glucose (Gibco Invitrogen, Grand Island, NY, USA), SC in SC medium (ScienCell Research Laboratories, San Diego, CA, USA)) without or with 1 µM Sorafenib (Nexavar, Bayer, Leverkusen, Germany) for 24 h. Supernatants (=conditioned medium) were centrifuged, aliquoted and stored at −20 °C for analyzing the impact of CAF and SC on signaling pathways in CCA cell lines.

### 4.6. Western Blot to Analyze Activation of Signaling Pathways

To analyze the impact of CAF and SC on distinct signaling pathways in the CCA cell lines, 3 × 10^5^ HuCCT-1 and TFK-1 cells were seeded per well in a 6-well plate. The following day, the medium was removed, cells were washed twice with PBS and cells were cultured in culture medium of CAF (DMEM) or SC (SC medium) without FCS for 24 h.

The next day, cells were either left untreated or treated with 1 µM Sorafenib. After another 24 h, the medium was removed and cells were cultured for 15 min in conditioned medium from CAF or SC (see above) supplemented with or without 1 µM Sorafenib. As the control, cells were cultured in culture medium without FCS and without preconditioning by SC and CAF to rule out a direct inhibition of the signaling pathways in the cancer cells by Sorafenib.

Cell lysates were generated from the differentially cultured cells by lysing the cells in 80 µL RIPA buffer inclusive PhosSTOP and Complete Protease Inhibitor Cocktail (both Roche, Penzberg, Germany) according to established protocols [57]. SDS-gel electrophoresis as well as Western blotting was conducted as described previously [56] using the following antibodies (all Cell Signalling Technology, Leiden, Holland; diluted in BSA, Carl Roth GmbH, Karlsruhe, Germany): STAT3 (mouse 124H6, 1:1000, 5% BSA in TBS-T), Phospho-STAT3 (rabbit 1:300, 5% BSA in TBS-T), SAPK/JNK (both rabbit 1:500, 0.5% BSA in TBS-T) and Phospho-SAPK/JNK antibody (rabbit 1:500, 0.5% BSA in TBS-T), AKT (mouse, 1:1000, 5% BSA in TBS-T) Phospho-AKT (rabbit 1:1000, 5% BSA in TBS-T), p44/42 MAPK (ERK1/2) (rabbit 1:1000, 5% BSA in TBS-T) and Phospho-p44/42 MAPK (ERK1/2) (rabbit 1:300, 5% BSA in TBS-T). ß-actin was used in a concentration of 1:10,000 (Sigma Aldrich Chemie GmbH, Steinheim, Germany, 0.5% milk in TBS-T).

### 4.7. Statistical Analyses

Statistical analyses were carried out using SPSS 23.0 (SPSS, IBM Corporation, Armonk, NY, USA). Represented are mean values ± standard deviation. The data were tested for normal distribution and equal variances using the Shapiro–Wilk test. Non-parametric datasets of different groups were analyzed using the Kruskal–Wallis one-way ANOVA on ranks. Statistical significance was defined at a *p*-value of <0.05, according to the Dunn method for non-parametric data. Survival analyses were performed by Kaplan–Meier estimates and statistical evaluations were performed using log-rank tests. Significances were defined using asterisks: * *p* < 0.05, ** *p* < 0.01, *** *p* < 0.001.

## 5. Conclusions

Overall, this study contributes toward a better understanding of the paracrine interplay of CAF and SC, respectively, with CCA cells and provides explanations on how these cells may impact cell migration, with this being an important metastasis-associated process. Co-cultivation of the two CCA cell lines HuCCT-1 and TFK-1 with CAF increased tumor cell migration and secretion of the pro-inflammatory cytokines MCP-1, CXCL-1, IL-6 and IL-8, which could be clearly diminished by Sorafenib treatment. In contrast, co-culture of CCA cells with SC predominantly increased the cell migration of SC, which could not be inhibited by Sorafenib. The fact that Sorafenib treatment inhibited the cell migration of CAF and CAF co-cultured CCA but not those of SC, along with the observation that signaling pathways that are important for malignant progression are also not impacted by Sorafenib in both CCA cell lines, underscores that the therapeutic efficacy of Sorafenib is highly context-dependent and provides an explanation as to why Sorafenib failed in the treatment of CCA patients in some clinical studies [25,26]. Thus, further studies are needed to elucidate the conditions determining the efficacy of Sorafenib for treatment of CCA.

## Figures and Tables

**Figure 1 jcm-11-02785-f001:**
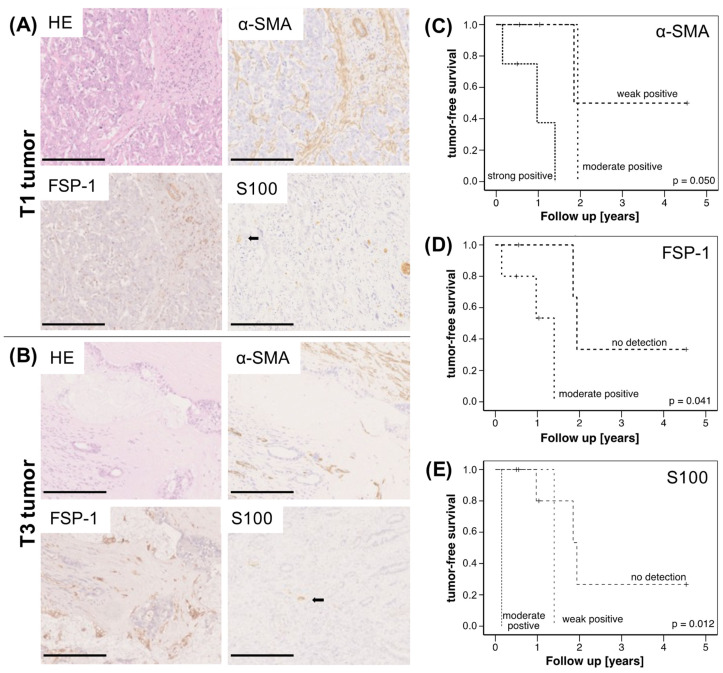
Immunohistochemical analysis of CCA specimens and correlation with survival of CCA patients. (**A**) Representative HE stainings and stainings for α-SMA, FSP-1 and S100 in pT1 N0 (scale bar = 250 µm) and (**B**) pT3 N0 (scale bar = 500 µm) CCA specimens. Kaplan–Meier plots for tumor-free survival of CCA patients depending on (**C**) α-SMA, (**D**) FSP-1- and (**E**) S100-expression; (*n* = 14 CCA, thereof 5 with intrahepatic and 9 with extrahepatic tumors). Statistical significances are indicated in the figure.

**Figure 2 jcm-11-02785-f002:**
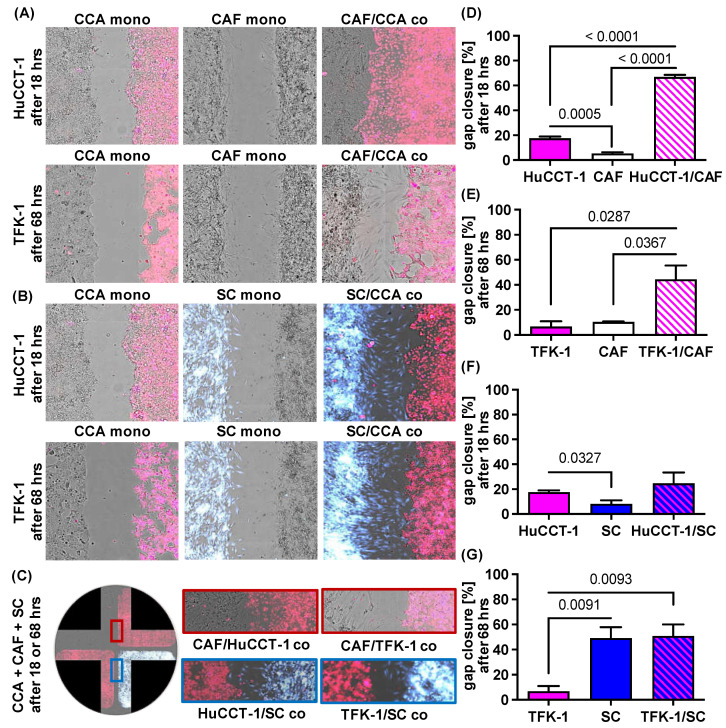
Paracrine impact of CAF and SC on cell migration of CCA cells. HuCCT-1 and TFK-1 cells were cultured either alone (CCA mono) or together with CAF (CAF/CCA co) or SC (SC/CCA co) in 4 chamber ibidi slides. Mono-cultured CAF (CAF mono) and SC (SC mono) were cultured in parallel as control. Mono- and co-cultures with HuCCT-1 cells were analyzed after 18 h and those of TFK-1 cells after 18 h and 68 h (representative pictures at 10-fold magnification). CCA cells were stained with CellTracker Deep Red, SC were stained with CellTracker CMAC blue and CAF were not stained. (**A**,**B**) Representative pictures of gap closure of the different mono- and co-cultured cell populations. (**C**) Representative images of co-cultured CCA cells, CAF and SC in a 4 chamber ibidi slide showing gap closure after 18 h (HuCCT-1) and 68 h (TFK-1). (**D**–**G**) Quantification of cell migration of the depicted cell populations after 18 h (HuCCT-1) and 68 h (TFK-1) were performed with the Wound healing beta 2F-Operator (Synentec). Data are presented as % gap closure normalized to t = 0 h and as mean and standard deviation of 3 independent experiments. Statistical significances are indicated in the figure.

**Figure 3 jcm-11-02785-f003:**
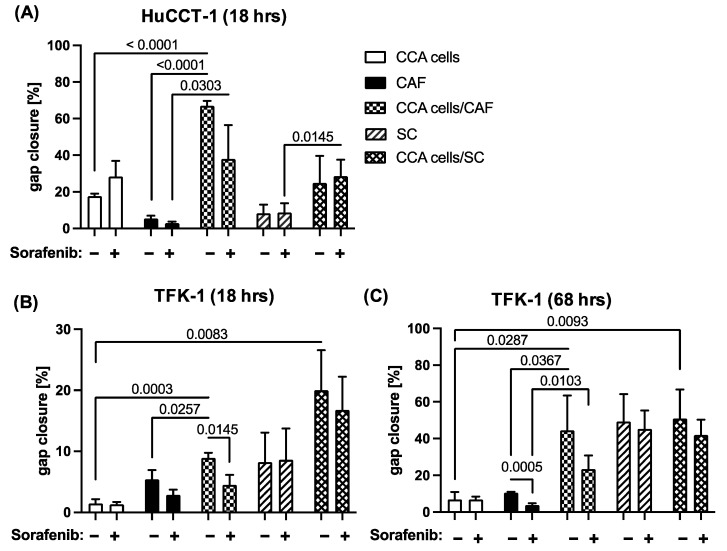
Sorafenib reduces migration of CAF and CAF co-cultivated CCA cells. HuCCT-1 and TFK-1 cells were cultured either alone or together with CAF or SC in 4 chamber ibidi slides. Mono-cultured CAF and SC were cultured in parallel as control. Additionally, cells were either left untreated or treated with 1 µM Sorafenib. (**A**) Gap closure of mono- and co-cultures with HuCCT-1 cells was analyzed after 18 h and those with TFK-1 cells after (**B**) 18 h and (**C**) 68 h. Data are presented as % gap closure normalized to t = 0 h and as mean and standard deviation of 3 independent experiments. Statistical significances are indicated in the figure.

**Figure 4 jcm-11-02785-f004:**
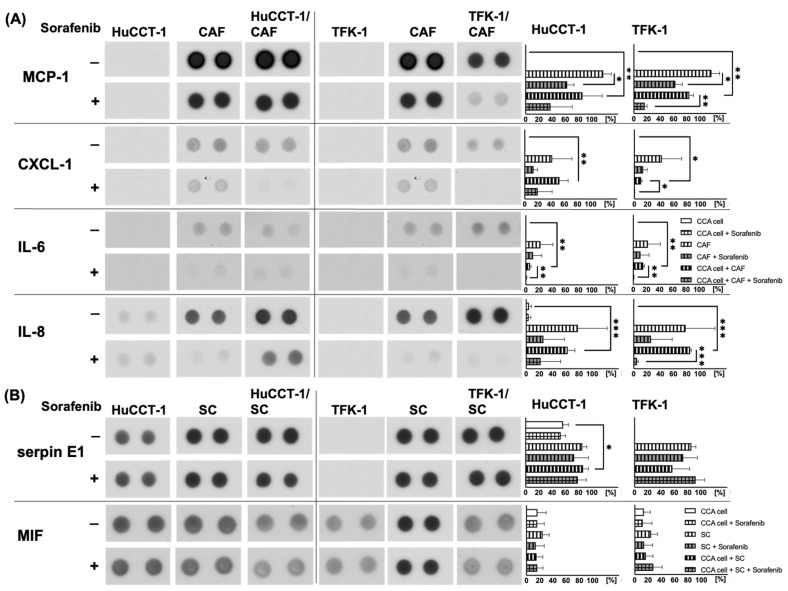
Detection of human cytokines and chemokines in supernatants of mono- and co-cultured CCA and stromal cells. HuCCT-1 and TFK-1 cells were cultured either alone or together with CAF or SC in transwell systems. Mono-cultured CAF and SC were cultured in parallel as control. Additionally, cells were either left untreated or treated with 1 µM Sorafenib for 72 h. Cell culture supernatants were analyzed using the Proteome Profiler Human Cytokine Array Kit, Panel A. Shown are representative results of mono- and co-cultures of CCA cells with (**A**) CAF and (**B**) SC. Pictures are taken from one experiment out of three independent biological replicates. Right to each blot, mean spot densitometries of all replicates ± standard deviation are shown. Significances were defined using asterisks: * *p* < 0.05, ** *p* < 0.01, *** *p* < 0.001.

**Figure 5 jcm-11-02785-f005:**
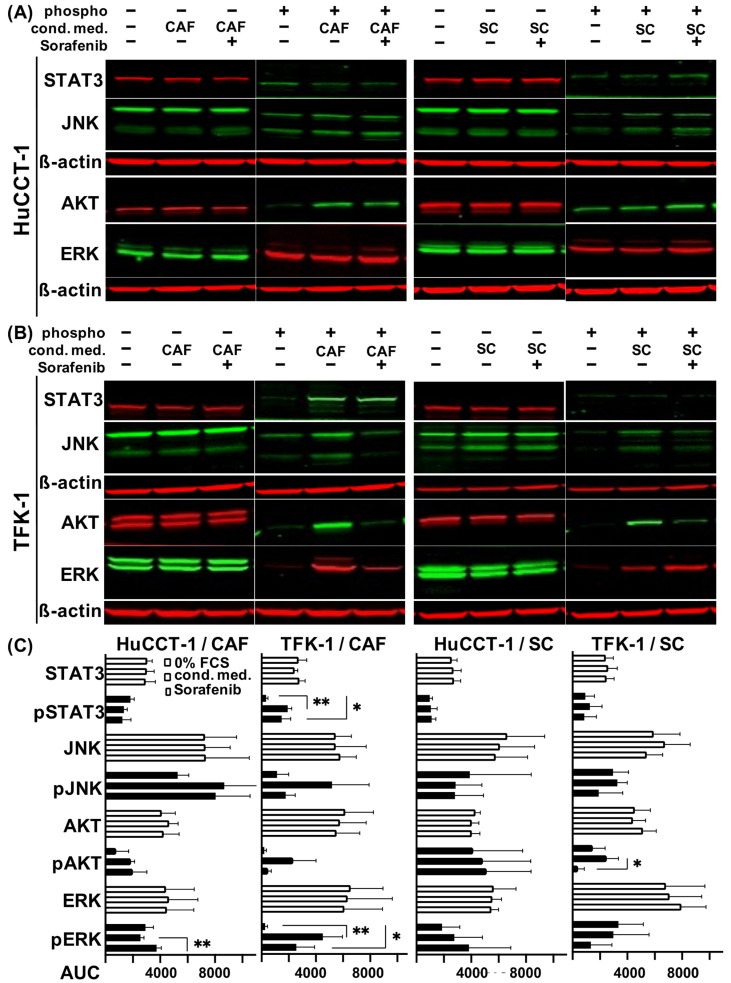
Impact of CAF and SC on signaling pathways in CCA cells in the absence or presence of Sorafenib. (**A**) HuCCT-1 cells and (**B**) TFK-1 cells were initially cultured in 0% FCS medium for 24 h with additional Sorafenib treatment for another 24 h where appropriate. The following day, cells were treated for 15 min in unconditioned medium or in conditioned medium from CAF or SC, which was either left untreated or supplemented with 1 µM Sorafenib beforehand. Shown are representative blots out of three replicates demonstrating expression levels of the phosphorylated and total forms of STAT3, JNK, AKT and ERK. ß-actin was determined in parallel as control. (**C**) Quantifications of fluorescence signals of the detected total and phosphorylated signaling proteins are demonstrated as area under the curve (AUC). Bars represent from top to bottom unconditioned medium, treatment with conditioned medium and Sorafenib treatment. Significances were defined using asterisks: * *p* < 0.05, ** *p* < 0.01.

## Data Availability

The clinical datasets supporting the conclusions of this study were derived from patient files (paper and electronic form). Therefore, restrictions to availability apply due to data protection regulations. Anonymized data are, however, available from the corresponding author on reasonable request and with permission of the University Hospital Schleswig-Holstein and the local review board.

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
