# Peer review of "Paracrine Interaction of Cholangiocellular Carcinoma with Cancer-Associated Fibroblasts and Schwann Cells Impact Cell Migration"

_jcm, 2022, doi:10.3390/jcm11102785_

Round 1
Reviewer 1 Report
Minor comments:
- In Figure 2D, CAF and HuCCT-1 are too close.
- The figure pattern of HuCCT-1/SC in Figure 2F and TFK-1/SC in Figure 2G is not consistent.
- The asterisks to show the significance in Figure 4 are missing.
- The significance values to show the effects of Sorafenib treatment on CCA cells/CAF are missing.
- The treatment of Sorafenib can affect the secretion of MCP-2, CXCL-2, IL-6, and IL-8 as shown in Figure 4, how about the effect of corresponding antibodies of those cytokines and chemokines on the migration of CAF and CAF co-culture CCA cells?
Author Response
Reviewer 1’s comments:
Minor comments:
- In Figure 2D, CAF and HuCCT-1 are too close.
Answer: Thank you for this advice, we have adapted it accordingly.
- The figure pattern of HuCCT-1/SC in Figure 2F and TFK-1/SC in Figure 2G is not consistent.
Answer: Thank you for this advice, we have adapted it accordingly.
- The asterisks to show the significance in Figure 4 are missing.
Answer: We thank the reviewer for this important advice. We have erased this formatting difficulty.
- The significance values to show the effects of Sorafenib treatment on CCA cells/CAF are missing.
Answer: Thank you for this advice, however, we already wrote in our manuscript in § 2.3. Sorafenib inhibits migration of CCA and stromal cells in a context dependent manner.: “Furthermore, clear reduction of cell migration was observed in co-cultures from HuCCT-1 cells and CAF (from 66.9 % to 38.0 %, p = 0.0548). … In line with the results described for HuCCT-1 cells and mono-cultured CAF, a clear inhibitory effect on cell migration could only be observed on co-cultures of CAF and TFK-1 cells leading to a reduced gap closure from 8.9 % to 4.5 % (p = 0.0145) and from 44.5 % to 23.4 % (p = 0.1489) after 18 hrs and 68 hrs, respectively (Figure 3B+C).” For reasons of clarity, we have only demonstrated statistically significant values in the graph.
- The treatment of Sorafenib can affect the secretion of MCP-2, CXCL-2, IL-6, and IL-8 as shown in Figure 4, how about the effect of corresponding antibodies of those cytokines and chemokines on the migration of CAF and CAF co-culture CCA cells?
Answer: We thank the reviewer for this valuable advice. Our aim of this study was to identify cytokines and chemokines that are modified by the paracrine interaction between CCA cells and either of the two different stromal cell populations. Of note, we could demonstrate that certain factors are already detectable at elevated levels in supernatants of mono-cultured stromal cells so that it can be speculated whether these factors are responsible for the observed effects on the migratory behavior of CCA cells. However, since the elaboration on whether these factors are causative for the observed functional effects of the CCA cells requires a substantial amount of time, we wish to combine these analyses with further studies on this topic in an upcoming manuscript. We have therefore taken up the suggestion by the reviewer and specified this in the discussion section: “Thus, future studies involving experiments with blocking antibodies will have to examine whether these inflammatory factors (such as MCP-1, CXCL-1, IL-6 or IL-8) are responsible for enhanced migration of CCA cells and SC, respectively.”

Reviewer 2 Report
Gundlach et al investigated the impact of CAFs and Schwann cells on CCA cancer cells and how multityrosine kinase-inhibitor Sorafenib affects the interaction. They found that the enhanced migration ability of CCA cells in co-culture with CAFs as well as pro-inflammatory cytokines secretion could be reversed by Sorafenib treatment, while Schwann cells impact CCA cells differently and Sorafenib didn’t affect the migration in co-culture. This study replenished the understanding of the paracrine interaction of cancer cells and stomal cells, which has been attracting a lot of interest recently. However, the manuscript still needs major improvement before further consideration.
Major comments:
- In general, the manuscript was written mostly with description of experiments results without enough explanation and discussion. The manuscript could benefit with discussing the results instead of just listing the observations in detail.
- CAFs population have been demonstrated to be heterogenous and classified into multiple subtypes. The functions of different CAF populations can be largely different. The phenotype of CAF used in this manuscript needs further characterization, such as surface markers expression, function and which subtype these CAFs fall into.
- At last, the authors concluded that CAF and SC impact signaling pathways in CCA cells in a paracrine manner. However, only one cell line supported the conclusion. In general, the authors mainly used two cell lines in the whole study, which is not enough given the heterogeneity. More CCA cell lines should be included to support the conclusions.
Minor comments:
- In Figure 2, the time phase of each panel should be labeled.
- In Figure 4, the legends of bar graphs and statistical analysis label were confusing. There were no * in the bar graphs.
- In line 216, “SC was well as in co-cultures of both CCA” should be “SC as well as in co-cultures of both CCA”.
- In Figure 5, the data of direct effects of Sorafenib on signaling pathways of CCA cancer cells should be included, which are important controls. The authors just described the findings without data.
Author Response
Reviewer 2’s comments
Gundlach et al investigated the impact of CAFs and Schwann cells on CCA cancer cells and how multityrosine kinase-inhibitor Sorafenib affects the interaction. They found that the enhanced migration ability of CCA cells in co-culture with CAFs as well as pro-inflammatory cytokines secretion could be reversed by Sorafenib treatment, while Schwann cells impact CCA cells differently and Sorafenib didn’t affect the migration in co-culture. This study replenished the understanding of the paracrine interaction of cancer cells and stomal cells, which has been attracting a lot of interest recently. However, the manuscript still needs major improvement before further consideration.
Major comments:
- In general, the manuscript was written mostly with description of experiments results without enough explanation and discussion. The manuscript could benefit with discussing the results instead of just listing the observations in detail.
Answer: We thank the reviewer for this remark and kindly refer to the discussion section. Our study aimed at elucidating the paracrine impact of CAF and SC on CCA cells with focus on cell migration and whether this can be targeted by Sorafenib. We have therefore considered our results in the context of other studies and discussed them thoroughly. Furthermore, we outline topics that should be investigated in the future with a particular focus on the clinical impact:
…”In line with other studies, the results of our study showed lower survival rates in CCA patients with an increased immunohistochemical staining of a-SMA in CAF in surgically resected intrahepatic CCA [16, 53, 54]. Prospective risk assessment regarding survival after surgery could favor patients with a low density of a-SMA in CAF. Therefore, tumor biopsies in the evaluation process before surgery could add to a better selection of patients, who might benefit from surgery with a better tumor-free survival for CCA with a low density of a-SMA in CAF. Furthermore, the results of the study suggest that a suppression of the crosstalk between CAF and CCA cells leads to an impaired tumor invasion. Therefore, future research should focus to identify targets by which deregulated expression and release of tumor stimulating cytokines by CAF is reversed. For extrahepatic CCA, a migration of SC towards the tumor cells was seen. Extrahepatic CCA cells need SC for perineural invasion to invade in the liver [21]. Since the results of this study demonstrate migration of SCs towards CCA cells and a synergistic effect of SC and CCA cells in co-culture, future research should investigate the mechanisms of the migration of SC towards CCA cells in more detail and targets should be identified to block this malignancy promoting crosstalk.”
- CAFs population have been demonstrated to be heterogenous and classified into multiple subtypes. The functions of different CAF populations can be largely different. The phenotype of CAF used in this manuscript needs further characterization, such as surface markers expression, function and which subtype these CAFs fall into.
Answer: We thank the reviewer for his comment. The CAF generation used in our manuscript has been described in methods section 4.2. Our CAFs were prepared from two different human CCA tissues and pooled after in vitro selection. For the determination of the CAF, we referred to a previous study by our working group, in which the presence of CAF was confirmed using immunohistochemical parameters: “CAF were prepared from two different human CCA tissues and pooled after in vitro selection as previously described and tested by positive staining for a- SMA and Vimentin as well as negative staining for the pan-cytokeratin marker KL-1 [56]. Patients gave their consent for use of their tumor tissue and the procedure was approved by the ethics committee of Kiel University (A 110/99). In brief, immediately after resection, …”. Nevertheless, we agree with the reviewer that CAF heterogeneity is emerging in recent years, especially for pancreatic and breast cancers. However, little is known about CAF in CCA and nothing is known about CAF heterogeneity in CCA, thus representing an interesting research area for upcoming studies. We have therefore included this important issue in our discussion as follows: “Of note, since CAF have been demonstrated to be heterogeneous in different other tumor entities, especially in pancreatic and breast cancer [13], future research should also focus on CAF heterogeneity in CCA in order to elucidate whether different CAF subtypes might exert pro- and antitumorigenic effects in CCA cells.” 3. At last, the authors concluded that CAF and SC impact signaling pathways in CCA cells in a paracrine manner. However, only one cell line supported the conclusion. In general, the authors mainly used two cell lines in the whole study, which is not enough given the heterogeneity. More CCA cell lines should be included to support the conclusions.
Answer: Being aware of that the observed effects were not consistently detectable in the two CCA cell lines used, we have taken care throughout the manuscript to refer to an association or to suggest that factors may play a role. Further studies are needed to elucidate the underlying relationships, but these are beyond the scope of this study and are therefore planned (as are other investigations) in a separate study. As part of this, the use of additional cell lines is also planned. We have added this issue in the discussion: “Moreover, owing to the fact that our results are based on the use of two CCA cell lines that differ in their origin (intrahepatic vs. extrahepatic) and therefore generalization is difficult in the presence of inconsistent findings observed, the use of additional CCA cell lines as well as CAF populations is planned for further studies.”
Minor comments:
- In Figure 2, the time phase of each panel should be labeled.
Answer: We thank the reviewer for this remark. This information has been added accordingly.
- In Figure 4, the legends of bar graphs and statistical analysis label were confusing. There were no * in the bar graphs.
Answer: We thank the reviewer for this comment. We have erased this formatting difficulty.
- In line 216, “SC was well as in co-cultures of both CCA” should be “SC as well as in co- cultures of both CCA”.
Answer: We thank the reviewer for this remark and have corrected this mistake.
- In Figure 5, the data of direct effects of Sorafenib on signaling pathways of CCA cancer cells should be included, which are important controls. The authors just described the findings without data.
Answer: We thank the reviewer for pointing this out. We have adapted our results accordingly: “For the TFK-1 cells a light activation of ERK and JNK was observed after Sorafenib treatment (AUC in phospho-ERK from 201 to 8490 and in phospho-JNK from 331 to 5941). This finding in the mono-culture contrasted to the inhibition of the different pathways after Sorafenib treatment of the cancer cell-SC and -CAF co-cultures. For the AKT and JAK-STAT pathway no difference in activation was observed after treatment with Sorafenib. In mono-cultured HuCCT- 1 cells no difference in activation of the different pathways was observed after treatment with Sorafenib either (phospho-STAT3 from 6625 to 5992; phospho-JNK from 7925 to 7627; phospho-AKT from 5278 to 2302 and phospho-ERK from 9109 to 8432 AUC, respectively).”

Round 2
Reviewer 2 Report
The figure 2 was overlapped with two layers and the time phase of each panel is still missing in the figure.
Author Response
Dear reviewer,
we thank for your kind advices and the amount of time you spent processing it. We have erased the double-layer bug, which has been caused by the follow-track function. Furthermore, we have added the time phase for both, figures and diagrams. We very much hope, that these improvements meet your approval.
Kind regards
Jan-Paul Gundlach